# Chloride dynamics alter the input-output properties of neurons

**Christopher B. Currin**[1], **Andrew J. Trevelyan**[2], **Colin J. Akerman**[3], **Joseph V. Raimondo**[1]*

**1** Division of Cell Biology, Department of Human Biology, Neuroscience Institute and Institute of Infectious Disease and Molecular Medicine, Faculty of Health Sciences, University of Cape Town, Cape Town, South Africa, **2** Institute of Neuroscience, Newcastle University, Newcastle upon Tyne, United Kingdom, **3** Department of Pharmacology, University of Oxford, Oxford, United Kingdom

* joseph.raimondo@uct.ac.za

**Data Availability Statement:** All the experimental data presented in this manuscript has been made publicly available and may be accessed using the following link: http://raimondolab.com/2020/05/07/chloride-dynamics-data/. All the code to generate the modelling data has been made publicly

## Abstract

Fast synaptic inhibition is a critical determinant of neuronal output, with subcellular targeting of synaptic inhibition able to exert different transformations of the neuronal input-output function. At the receptor level, synaptic inhibition is primarily mediated by chloride-permeable Type A GABA receptors. Consequently, dynamics in the neuronal chloride concentration can alter the functional properties of inhibitory synapses. How differences in the spatial targeting of inhibitory synapses interact with intracellular chloride dynamics to modulate the input-output function of neurons is not well understood. To address this, we developed computational models of multi-compartment neurons that incorporate experimentally parametrised mechanisms to account for neuronal chloride influx, diffusion, and extrusion. We found that synaptic input (either excitatory, inhibitory, or both) can lead to subcellular variations in chloride concentration, despite a uniform distribution of chloride extrusion mechanisms. Accounting for chloride changes resulted in substantial alterations in the neuronal input-output function. This was particularly the case for peripherally targeted dendritic inhibition where dynamic chloride compromised the ability of inhibition to offset neuronal input-output curves. Our simulations revealed that progressive changes in chloride concentration mean that the neuronal input-output function is not static but varies significantly as a function of the duration of synaptic drive. Finally, we found that the observed effects of dynamic chloride on neuronal output were mediated by changes in the dendritic reversal potential for GABA. Our findings provide a framework for understanding the computational effects of chloride dynamics on dendritically targeted synaptic inhibition.

## Author summary

The fundamental unit of computation in the brain is the neuron, whose output reflects information within the brain. A determining factor in the transfer and processing of information in the brain is the modulation of activity by inhibitory synaptic inputs. Fast synaptic inhibition is mediated by the neurotransmitter GABA binding to GABA$_A$ receptors, which are permeable to chloride ions. How changes in chloride ion concentration affect

available and may be accessed using the following link: https://github.com/ChrisCurrin/chloride-dynamics-io-neuron.

**Funding:** CBC is supported by the German Deutscher Akademischer Austauschdienst (DAAD, https://www.daad.de/), the South African (SA) National Research Foundation (NRF, https://www.nrf.ac.za), and the University of Cape Town (UCT, https://www.uct.ac.za). AJT is supported by a grant from the Biotechnology and Biological Sciences Research Council (BBSRC, https://bbsrc.ukri.org) (BB/P019854/1) and the (MRC, https://mrc.ukri.org/) (MR/R005427/1). Some of the research leading to these results has received funding from European Research Council (ERC, https://erc.europa.eu) grant agreement number 617670 to CJA. JVR has received funding support from the Blue Brain Project (https://www.epfl.ch/research/domains/bluebrain/), the SA NRF, SA Medical Research Council (MRC, http://www.mrc.ac.za), Wellcome Trust (https://wellcome.ac.uk) and the Future Leaders – African Independent Research (FLAIR, https://royalsociety.org/grants-schemes-awards/grants/flair/) Fellowship Programme (FLR \R1\190829): a partnership between the African Academy of Sciences and the Royal Society funded by the UK Government's Global Challenges Research Fund. The funders had no role in study design, data collection and analysis, decision to publish, or preparation of the manuscript.

**Competing interests:** The authors have declared that no competing interests exist.

neuronal output is an important consideration for information flow in the brain that is currently not being thoroughly investigated. In this research, we used multi-compartmental models of neurons to link the deleterious effects that accumulation of chloride ions can have on inhibitory signalling with changes in neuronal ouput. Together, our results highlight the importance of accounting for changes in chloride concentration in theoretical and computer-based models that seek to explore the computational properties of inhibition.

## Introduction

Neurons in the brain communicate with one another via synaptic signalling, which relies on the activation of receptor proteins that permit rapid transmembrane fluxes of ions. A neuron's principal operation is to transform synaptic inputs into action potentials. The rate at which action potentials are generated, a neuron's output firing rate, is a primary means by which neurons encode information [1]. The manner in which a neuron integrates rate-coded synaptic input, and transforms it into an output firing rate, is referred to as its input-output function [2].

Synaptic inhibition is crucial for shaping this input-output transformation [3,4]. For example, inhibitory inputs can perform a divisive operation by reducing the slope (gain) of the input-output function, or a subtractive operation by offsetting the input-output function to the right [5,6]. Previous work has demonstrated that this differential algebraic effect of synaptic inhibition can be realised through targeting inhibitory inputs to different areas of the pyramidal cell. Proximally located inhibition close to the soma can have a divisive effect on the input-output function whilst spatially distributed, dendritically targeted inhibition supports subtractive effects of inhibition [7].

At the receptor level, synaptic inhibition in the brain is primarily mediated by type A γ-aminobutyric acid receptors (GABA$_A$Rs), which are selectively permeable to chloride ions (Cl$^-$) and, to a lesser extent, bicarbonate ions (HCO$_3^-$, see Table 1 for a list of abbreviations) [8]. As a result, the reversal potential for GABA$_A$Rs (EGABA), is largely a function of the transmembrane Cl$^-$ gradient. Together with the membrane potential, this sets the driving force for Cl$^-$ flux across these receptors and hence represents a fundamental property of inhibitory signaling [9]. Intracellular Cl$^-$ concentration ([Cl$^-$]$_i$), and hence EGABA, can differ between subcellular neuronal compartments, which has been suggested to result from spatial differences in the function or expression of cation-chloride cotransporters in the neuronal membrane [10–12]. That is, synaptic activation of GABA$_A$Rs can have different effects on the neuronal membrane potential because of differential expression of Cl$^-$ homeostatic mechanisms.

In addition to spatial differences in EGABA, inhibitory synapses also exhibit a form of short-term plasticity that involves changes in the ionic driving force for post-synaptic ionotropic receptors driven by short-term alterations in intracellular Cl$^-$ concentration [13,14]. Ionic plasticity occurs when GABA$_A$R-mediated Cl$^-$ influx overwhelms local Cl$^-$ extrusion via the canonical Cl$^-$ extruder KCC2 [15–17]. Previous work has shown how ionic plasticity is

**Table 1. Abbreviations.**

| | |
|---|---|
| Cl$^-$ | Chloride ion(s) |
| GABA$_A$R | Type A γ-aminobutyric acid receptor |
| HCO$_3^-$ | Bicarbonate ion(s) |

regulated on a spatial level with different neuronal subcellular compartments having differing susceptibility to activity-induced chloride accumulation. For example, both experimental and modelling studies have shown how small volume dendritic compartments are particularly prone to ionic plasticity [16,18–21].

Although the vast majority of theoretical models of inhibitory signalling and neuronal computation assume static values for EGABA, previous studies have explored the relevance of dynamic Cl$^-$ in multiple contexts. For example, studies have investigated the biophysical underpinnings of Cl$^-$ homeostasis via active ion transport, cation-chloride cotransporters, or impermeant anions [20,22], the effects of neuronal morphologies on Cl$^-$ accumulation [19,21,23], and how dynamic Cl$^-$ affects neural coding by degrading mutual information [24]. However, how differences in the spatial targeting of synaptic inhibition interact with ionic plasticity to dynamically modulate the input-output function of neurons is not well understood. To address this, we developed computational models of multi-compartment neurons, which incorporated experimentally parametrised mechanisms to account for neuronal Cl$^-$ extrusion and ionic plasticity. Firstly, we show that ongoing structured synaptic input (either excitatory, inhibitory, or both) can lead to spatial variations in EGABA independent of differences in Cl$^-$ extrusion. Secondly, we find that accounting for dynamic chloride has a significant effect on the ability of dendritically targeted inhibition to offset neuronal input-output curves. Thirdly, we demonstrate that due to ionic plasticity, the neuronal input-output function is not static but varies significantly as a function of time. Finally, we find that the observed effects of dynamic chloride on the neuronal output function are entirely mediated by changes in EGABA. Our results provide a framework for understanding how Cl$^-$ dynamics alter the computational properties of spatially targeted synaptic inhibition.

## Results

### Experimental characterisation of neuronal chloride extrusion

To explore the relevance of Cl$^-$ dynamics for the computational properties of synaptic inhibition, we developed multi-compartment models in NEURON that explicitly accounted for changes in [Cl$^-$]$_i$ across time and space. As a result, an important parameter within our model which we needed to determine was the rate of Cl$^-$ extrusion, which is mediated by the canonical Cl$^-$ cotransporter in mature neurons, KCC2 [25]. We utilised gramicidin perforated patch-clamp recordings in conjunction with optogenetic manipulation of [Cl$^-$]$_i$ to experimentally characterise Cl$^-$ extrusion in hippocampal neurons from mature rat organotypic brain slices (Fig 1A, schematic). Organotypic hippocampal brain slices possess mature Cl$^-$ homeostasis mechanisms, as evidenced by their hyperpolarising EGABA [26,27] and the fact that KCC2 is the major active Cl$^-$ transporter in these neurons [28]. Activation of the inward Cl$^-$ pump halorhodopsin from *Natronomonas pharaonis* (halorhodopsin, eNpHR3.0 or eNpHR) using a green laser (532 nm) for 15 s resulted in a profound increase in [Cl$^-$]$_i$. [Cl$^-$]$_i$ was determined by performing gramicidin whole-cell patch-clamp recordings (which do not disturb [Cl$^-$]$_i$) in conjunction with activation of GABA$_A$Rs using pressurised ejection of GABA (Fig 1A). This allowed [Cl$^-$]$_i$ to be calculated from EGABA measurements. We found that [Cl$^-$]$_i$ recovered to baseline levels over the course of approximately a minute following the 15 s eNpHR photocurrent. This recovery curve could be fitted using a single exponential function and replotted with Cl$^-$ extrusion rate (mM/s) as a function of Cl$^-$ concentration (Fig 1B). As Cl$^-$ extrusion by KCC2 is a function the transmembrane gradient for both K$^+$ and Cl$^-$, we modelled Cl$^-$ extrusion in the manner proposed by Fraser and Huang [29]:

$$V = P_{KCC2}([K^+]_i [Cl^-]_i - [K^+]_o [Cl^-]_o)$$

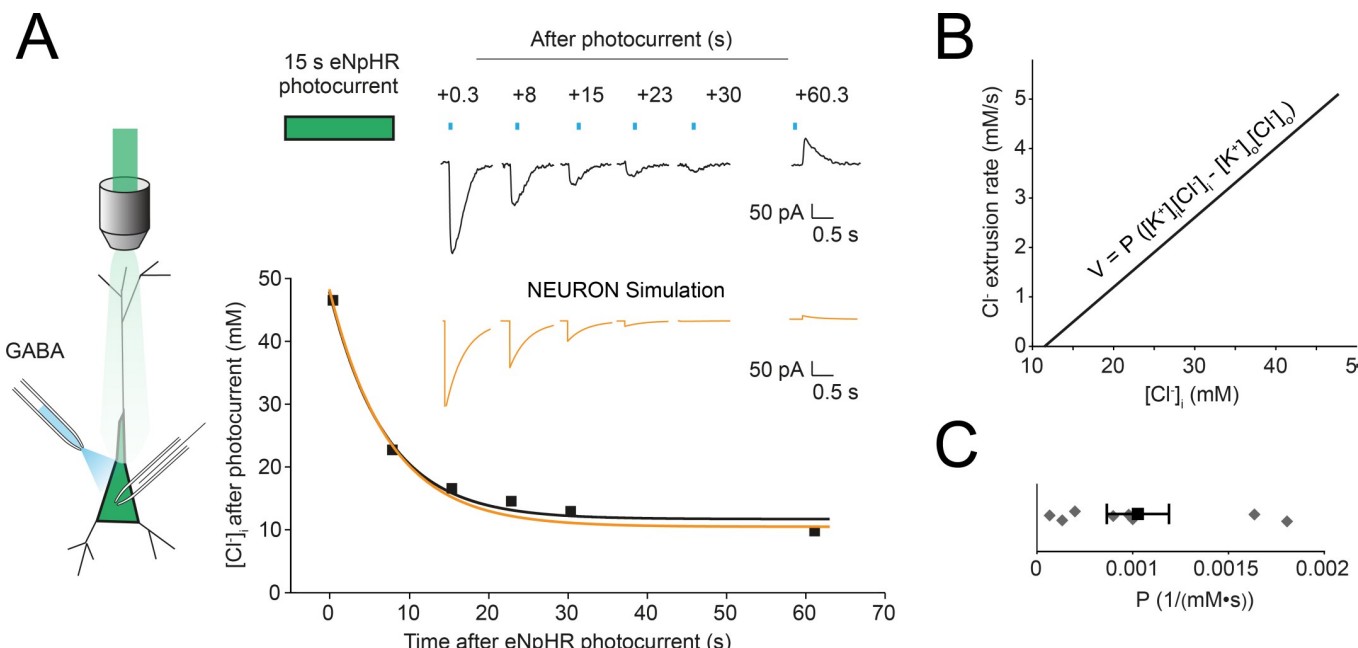

**Fig 1. Optogenetic quantification of chloride extrusion in hippocampal neurons.** (A) Left, schematic of the experimental setup where a gramicidin perforated patch is made from an eNpHR-expressing hippocampal pyramidal neuron. Green light is delivered via the objective and GABA puffs (blue) directed at the cell soma. Right, top, gramicidin perforated patch voltage-clamp recording from a neuron expressing eNpHR3.0-EYFP. GABA$_A$R currents recorded at different times, on different trials following 15s of Cl$^-$ load induced by light activation of eNpHR. Right bottom, EGABA and [Cl$^-$]$_i$ were calculated from each GABA$_A$R current (squares, see Methods) and plotted as a function of time after the photocurrent for a single cell. [Cl$^-$]$_i$ recovery was fitted by a single-exponential function (black). (B) From the data in 'A', KCC2 Cl$^-$ extrusion rate (V) as a function of [Cl$^-$]$_i$ was calculated. This allowed for the Cl$^-$ extrusion constant, P, to be calculated. (C) Population data from 8 neurons resulted in an average value of P of 0.001 mM$^{-1}$ · s$^{-1}$. A single compartmental model was then created using the NEURON simulation environment. By accounting for Cl$^-$ dynamics including Cl$^-$ extrusion via KCC2, a Cl$^-$ recovery curve and simulated GABA$_A$R currents could be generated (orange curves in 'A'). These closely match our experimental data.

where V is the Cl$^-$ extrusion rate and P$_{KCC2}$ is the Cl$^-$ extrusion constant, or "pump strength", of KCC2. The gradient of the line in Fig 1B can be used to calculate P$_{KCC2}$. We measured P$_{KCC2}$ as 1.0 ± 0.16 1/(M·s) (mean ± SEM, N = 8). P$_{KCC2}$ was reformulated as a function of surface area by assuming recorded pyramidal cells had a volume of 1.058 pL with a surface area of 529 x 10$^{-8}$ cm$^2$ [30] and current by multiplying by Faraday's constant, F, such that P$_{KCC2}$ = 1.9297 x 10$^{-5}$ mA/(mM$^2$·cm$^2$) (see S1 Text. Converting KCC2 "pump strength" parameter from $\frac{1}{mM \cdot s}$ to $\frac{mA}{mM^2 \cdot cm^2}$). This was then used to calculate the Cl$^-$ extrusion rate (V) for each compartment of our model at each point in time depending on the surface area of the compartment concerned and time-varying [Cl$^-$]$_i$. Combined with mechanisms to describe Cl$^-$ flux via both constitutively active Cl$^-$ leak channels and synaptic activation of Cl$^-$ permeable GABA$_A$Rs, as well as Cl$^-$ diffusion between compartments (see Methods), we were able to model how changes in [Cl$^-$]$_i$ both affects, and is affected by, spatially distributed patterns of synaptic drive.

## Spatial variation in intracellular chloride concentration is affected by the pattern of synaptic drive

In pyramidal neurons, synaptic excitation is predominantly located relatively far from the soma on the branches of the dendrites [31]. In contrast, synaptic inhibition either targets proximal structures (soma or proximal dendrites) or is co-located with excitation on peripheral dendrites [31]. Extensive evidence exists to suggest that different neuronal subcellular compartments (e.g. soma vs distal dendrites) can have different [Cl$^-$]$_i$. For example, experiments

using genetically encoded $Cl^-$ sensors [32], gramicidin perforated patch-clamping with photo-uncaging of GABA [33] or $Cl^-$ sensitive dyes [34] have shown different $[Cl^-]_i$ in the soma vs dendrites of neurons. It is therefore likely that GABAergic inputs to these compartments may have different functional effects [19–22]. Typically, these differences have been suggested to be driven by differences in the action of cation-chloride cotransporters such as KCC2 [10,11,35,36], however, it has also been well described that GABAergic input itself causes differential shifts in $[Cl^-]_i$ depending on the particular subcellular target location [16]. What has not been elucidated is how relative amounts of inhibitory and excitatory synaptic drive may drive spatial variations in $[Cl^-]_i$ despite uniform expression of cation-chloride cotransport.

To investigate this, we generated a multi-compartmental, conductance-based neuron with a short (50 μm), thick (2 μm), 'proximal' dendrite connected to a long (500 μm), thin (0.5 μm), 'distal' dendrite as in [7,37]. Synapses were modelled as conductance-based receptors receiving input at a particular input frequency (see Methods). Excitatory synapses were distally distributed, whilst inhibitory synapses were placed either distally (Fig 2A, inset) or proximally (Fig 2D, inset). Tonic "leak" currents for the three major ions ($K^+$:$Na^+$:$Cl^-$) with relative permeabilities of (1:0.23:0.4) established the steady-state membrane voltage (-71.35 mV) and resulted in an input resistance of 365 MΩ. In order to select a particular synaptic input configuration, the number of excitatory and inhibitory synapses used was determined by tuning the number of synapses, each receiving an input at a mean frequency of 5 Hz from independent Poisson processes, until the neuron produced an output firing rate of 5 Hz (white areas in Fig 2B) to simulate a network with average firing rates in the theta frequency range [38,39]. $[Cl^-]_i$ was initialized at 4.25 mM but allowed to evolve dynamically over the course of the 1 s simulation as a function of $Cl^-$ extrusion by KCC2 (uniformly distributed across all compartments), flux though $Cl^-$ leak channels, GABAergic synapses, and diffusion between compartments (see Methods). Note that multiple candidate sets of excitatory-inhibitory (E:I) synapses matched this objective (Fig 2B, blue squares).

Picking a synaptic configuration of 250 excitatory and 300 inhibitory synapses (250:300, E:I), the excitatory synapses alone (Fig 2A, red trace), inhibitory synapses alone (blue trace), or both combined (purple trace) were driven at 5 Hz and the final $[Cl^-]_i$ at the end of the 1 s simulation plotted as a function of distance from the soma (Fig 2A). We found that all three modes of synaptic drive produced shifts in $[Cl^-]_i$ in the dendrites relative to other subcellular compartments. Excitatory drive alone was sufficient to cause dendritic shifts in $[Cl^-]_i$ due to an increase in the driving force for $Cl^-$ influx via tonic $Cl^-$ currents (Fig 2A red trace), which in this case was similar to the $[Cl^-]_i$ shifts driven by inhibitory input alone (blue trace). Interestingly, given balanced input (equal amounts of excitatory and inhibitory input, purple trace), $[Cl^-]_i$ was elevated in the distal compartment more than the sum of the excitatory and inhibitory components. This could be accounted for by the shift in driving force for GABAergic input caused by the EPSPs, even at a low 5 Hz activity.

To further explore how synaptic drive affects the subcellular distribution of $[Cl^-]_i$, we utilised multiple sets of synaptic configurations (Fig 2C) as well as different frequencies of balanced input. This included higher frequencies of input to simulate short bursts of feedforward activity. The distal dendrite, receiving all the input, had the highest $[Cl^-]_i$ along the neuron, which was exacerbated by larger balanced input frequencies (50 Hz). This was true for all synapse pairs, but $[Cl^-]_i$ accumulation was most pronounced for 300:800 E:I synapses, given the greater number of GABAergic synapses (Fig 2C, right). These simulation results demonstrate that synaptic drive to the dendrites is an important factor in determining the subcellular distribution of $[Cl^-]_i$ within neurons.

As proximal dendrites tend to have larger diameters, and hence larger volumes we speculated that proximally targeted GABAergic input may have a differing effect on $[Cl^-]_i$. Relative

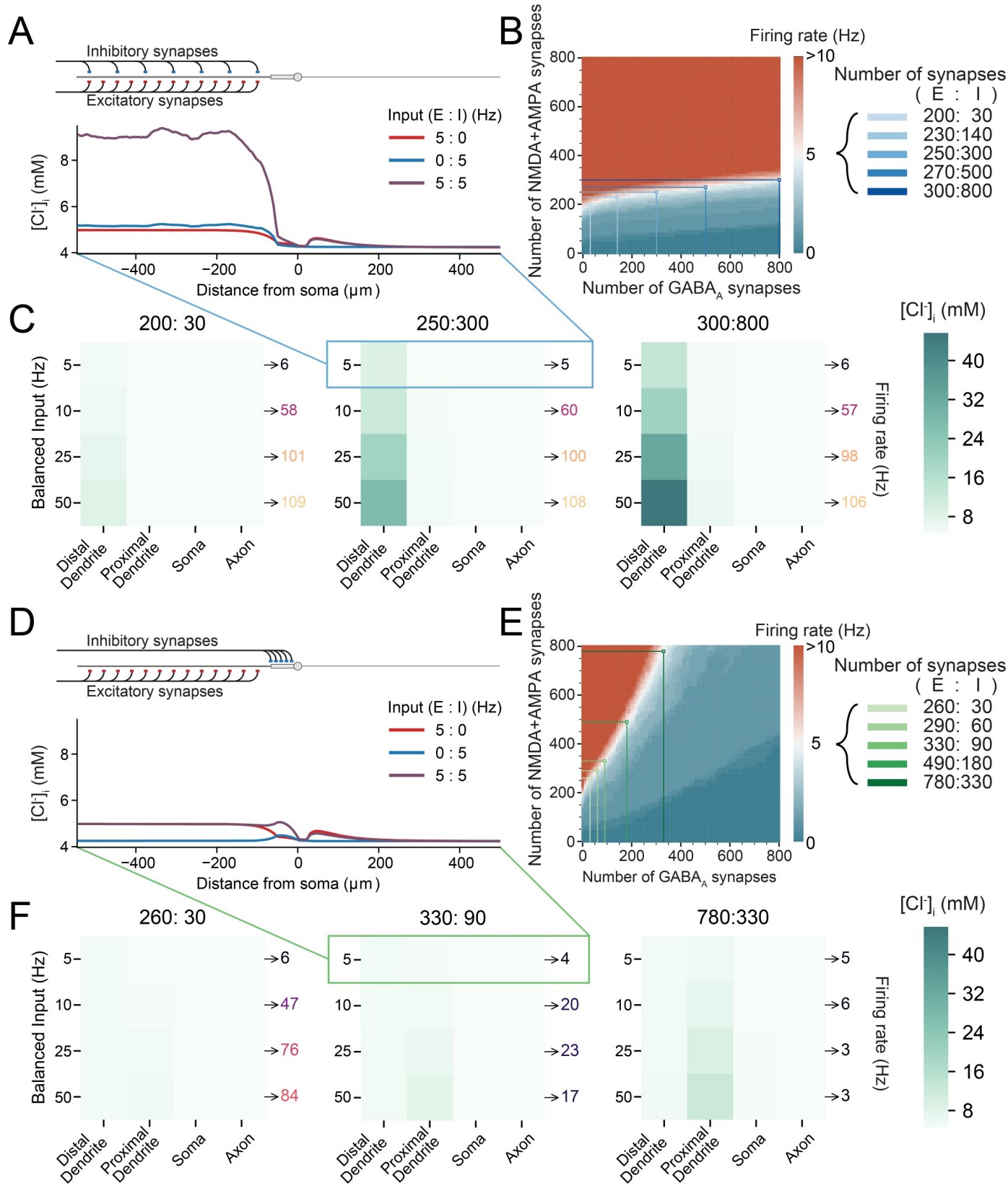

**Fig 2. The relative amounts of inhibitory and excitatory drive produce spatial variations in neuronal chloride concentration.** (A) Top, schematic depicting the model with an axon, soma, short thick 'proximal' dendrite connected to a long thin 'distal' dendrite. Both excitatory (red) and inhibitory (blue) synaptic input were evenly distributed across the distal dendrite. Bottom, $[Cl^-]_i$ concentration at the end of the 1 s simulations as a function of distance from the soma. Both inhibitory synaptic input alone (blue trace) and excitatory input alone (red trace) caused selective increases in $[Cl^-]_i$ in the distal dendrite. Balanced input (purple trace) caused an even greater increase in dendritic $[Cl^-]_i$. (B) Heatmap of neuronal firing rates as a function of different numbers of excitatory and inhibitory synapses used to select "balanced" synaptic configurations: i.e. pairs of excitatory-inhibitory (E:I) synapse numbers that could transform 5 Hz balanced input into 5 Hz output under conditions of dynamic $Cl^-$ (pale squares). (C) Heatmaps of $[Cl^-]_i$ (shades of green) in the distal dendrite, proximal dendrite, soma, and axon at the end of the 1 s simulations when $Cl^-$ was allowed to evolve dynamically for three pairs of synaptic numbers reflecting "weak" inhibition (left, E:I– 200:30), "moderate" inhibition (E:I– 250:300) and "strong" inhibition (E:I– 300:800) at different balanced input frequencies. Spatial variations in $[Cl^-]_i$ are dictated by the number of synapses and exaggerated by higher frequencies. (D) Top, schematic as in 'A' but with excitatory synapses targeted at the distal dendrite and the inhibitory synapses targeted at the proximal dendrite. Bottom, as in 'A', peri-somatic inhibitory synaptic input also produces spatial variations in $[Cl^-]_i$, but of smaller magnitude than when inhibitory synapses are distally located. (E) Heatmap as in 'B' but with proximal inhibition. This allowed E:I synapse pairs which produced 5 Hz output following 5 Hz input to be identified (pale squares). (F) While functionally similar to synapse pairs with distally targeted inhibition in 'C', pairs with proximal inhibition produced more modest changes in $[Cl^-]_i$ as a function of the subcellular domain.

amounts of excitatory and inhibitory synapses were chosen as before to maintain an output of 5 Hz following balanced 5 Hz input (Fig 2E, green squares over white). Inhibitory input at 5 Hz caused a small increase in $[Cl^-]_i$ in the proximal dendrite, with minor accumulation in the soma and early part of the distal dendrite due to diffusion (Fig 2D, blue). With excitatory input placed distally and inhibitory input placed proximally, their combined balanced input caused small accumulations of $[Cl^-]_i$ in the proximal dendrite. $[Cl^-]_i$ accumulation from proximal inhibition was lower overall given a similar number of inhibitory synapses and more excitatory synapses (260:30 proximal vs 200:30 distal, or 780:330 proximal vs 250:300 distal), despite the same frequencies of synaptic drive (Fig 2F).

Together these results demonstrate that despite a uniform subcellular distribution of $Cl^-$ extrusion by KCC2, spatial variations in $[Cl^-]_i$ can arise because of inhomogeneous distribution of synaptic drives across the dendrites. Furthermore, excitatory drive plays an important role in setting the neuron's $[Cl^-]_i$ by altering the driving force of $Cl^-$ flux through both tonic $Cl^-$ leak channels and $GABA_A$ synapses.

## Dynamic chloride accumulation compromises the effectiveness of inhibition during balanced distal synaptic input

Given the substantial effects of balanced synaptic drive on subcellular $[Cl^-]_i$ we observed in our simplified model of a pyramidal neuron, we next set out to determine how accounting for dynamic $Cl^-$ might affect neuronal output given differing frequencies of balanced synaptic drive. As in Fig 2B and 2E, different sets of excitatory and inhibitory synaptic configurations were chosen such that 5 Hz balanced input resulted in a 5 Hz output of the neuron under conditions of static $[Cl^-]_i$. This was done for both proximally targeted inhibitory synapses (green) and distally targeted inhibition (blue, Fig 3).

For neurons with proximally targeted inhibitory synapses (Fig 3A), the average output firing rate over the course of a 1 s simulation was recorded as a function of balanced synaptic input at multiple frequencies for different synaptic configurations (shades of green) and where $Cl^-$ was either allowed to evolve dynamically (solid line) or was held static (dashed lines). Although accounting for dynamic $Cl^-$ changed the exact timing of spikes compared to the static $Cl^-$ condition (Fig 3A, inset), the overall effect of dynamic $Cl^-$ on average firing rates was negligible at low frequencies (± 2 Hz difference at 10 Hz input), modest at medium frequencies (± 5 Hz at 20 Hz input), and only noticeably impactful at high frequencies for a small range of E:I synapse pairs (330:90 had ± 22 Hz difference, but 780:330 had 1 Hz difference).These overall small differences were likely due to the minimal changes in $[Cl^-]_i$ driven by proximally targeted inhibition as demonstrated in Fig 2E and the inset in Fig 3A.

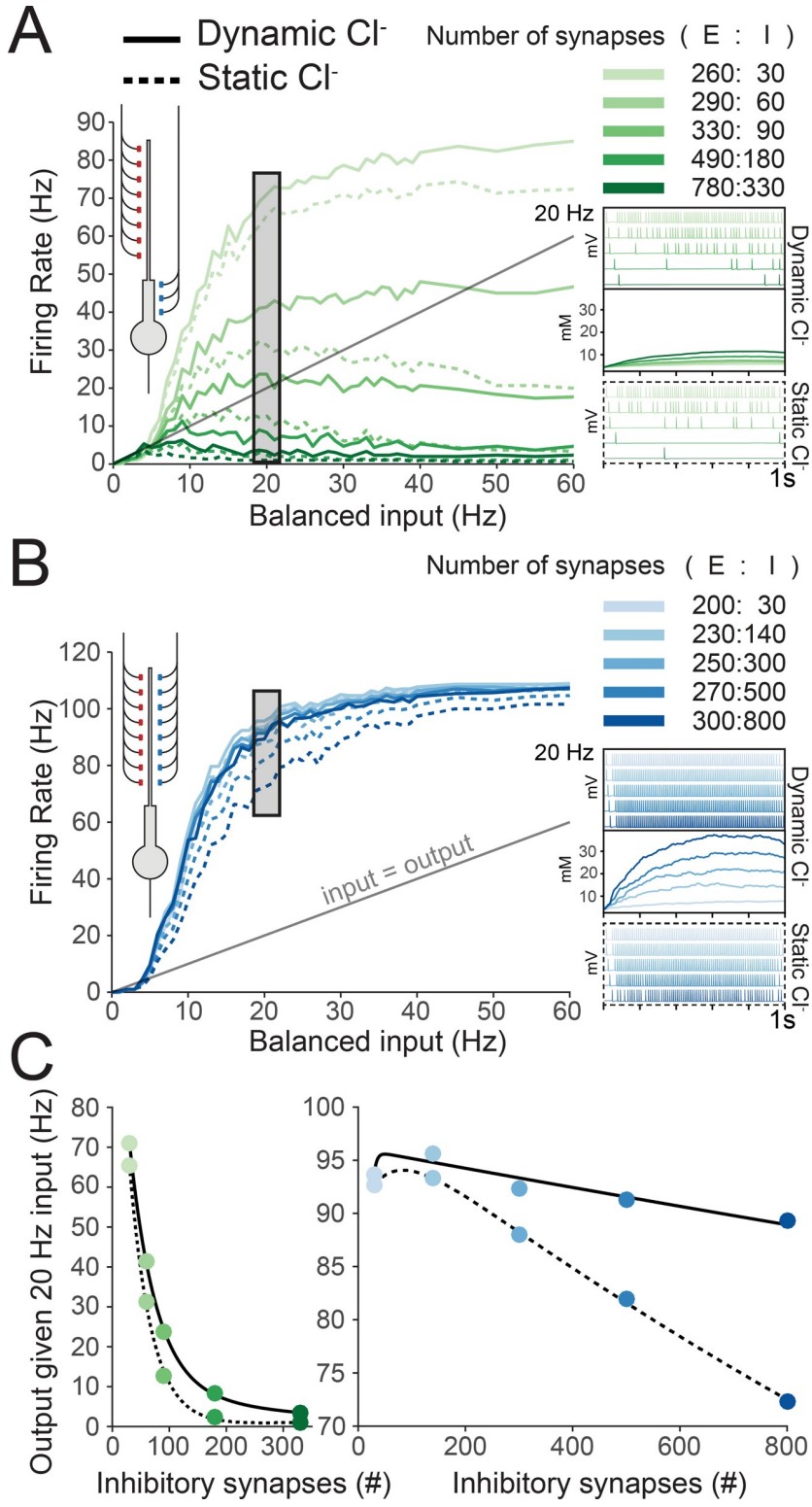

**Fig 3. Dynamic chloride accumulation compromises the effectiveness of inhibition during balanced distal synaptic input.** (A) Inner left, schematic of the model, depicting excitatory synaptic input targeted toward the distal dendrite and inhibitory synapses targeted at the proximal dendrite. Left, average output firing rates over the course of a 1 s simulation as a function of balanced synaptic input for different pairs of E:I synaptic numbers (shades of green, right). Each pair resulted in a 5 Hz output following 5 Hz balanced input (as in Fig 2). Simulations were performed

either with $[Cl^-]_i$ able to vary dynamically ("dynamic chloride", solid lines) or with $[Cl^-]_i$ held at a constant value ("static chloride", dashed lines). Insets with example voltage traces for simulation runs at 20 Hz input for both dynamic (top inset) and static (bottom inset) $Cl^-$ as well as $[Cl^-]_i$ for the dynamic $Cl^-$ simulations (middle inset). These show that accounting for $Cl^-$ dynamics results in obvious changes in spike timing. However, chloride dynamics did not result in large changes in output firing rates. (B) Inner left, schematic demonstrating excitatory and inhibitory inputs co-targeted toward the distal dendrite. Left, output firing rates following different balanced input frequencies with different pairs of E:I synaptic numbers (shades of blue, right) as in 'A', but with distally targeted inhibition. Dynamic $Cl^-$ resulted in large changes to output firing rates (solid vs dashed lines) as well as spike timing (inset, example simulation runs). (C) Output firing rate given 20 Hz balanced input for different numbers of inhibitory synapses targeted to the proximal dendrite (left) or the distal dendrite (right). Adding more inhibitory synapses in the case of distally targeted inhibtion did not meaningfully impact the firing rate when $Cl^-$ was dynamic. This was not the case for static $Cl^-$ where increasing the number of inhibitory synapses continued to decrease output.

In contrast, the results of the previous section suggest that balanced synaptic input with distally targeted inhibitory synapses has significant effects on dendritic $[Cl^-]_i$. To test the functional implications of this, we again measured the output firing rate during the 1 s simulation as a function of multiple balanced input frequencies and different synaptic configurations (shades of blue), but now with distally targeted inhibition either with $[Cl^-]_i$ allowed to evolve dynamically, or held static at its initial value (4.25 mM). We observed clear differences in both the timing and rates of action potential generation between simulations run with dynamic vs static $Cl^-$ (Fig 3B, inset). Dynamic $Cl^-$ resulted in higher output firing rates, particularly for medium balanced input frequencies (20 Hz, Fig 3B). Incorporating $Cl^-$ dynamics meant that for a wide range of balanced input frequencies, increasing the number of inhibitory synapses had a negligible effect on output firing rates (Fig 3C, right, solid line), with more GABAergic synapses having no additional "inhibitory" effect. That is, when $Cl^-$ is dynamic, inhibitory efficacy can not be recovered by simply increasing the number of inhibitory synapses. These data demonstrate that $Cl^-$ accumulation compromises the effectiveness of inhibition for controlling output firing rates in response to balanced synaptic input, especially when inhibition is distally targeted.

## Chloride dynamics differentially alter the neuronal input-output function based on inhibitory synaptic location

We next investigated how $Cl^-$ dynamics affect the ability of spatially targeted inhibition to alter the input-output function of a neuron. Previous work has demonstrated that dendritically located inhibition, by co-locating inhibition with excitation, can offset a neuron's input-output curve by increasing the threshold for firing [7,40,41]. In this manner, inhibition performs a subtractive operation on the neuronal output function [2]. In contrast, inhibition located more proximally than excitation can perform a divisive scaling of the neuronal output, by suppressing action potential generation [2,7,40]. How $Cl^-$ dynamics affect these operations however is unknown.

To explore this issue, we used the same morphologically simplified model of a pyramidal neuron as before (Fig 4A) to generate a classic input-output curve by plotting average neuronal firing rate during a 1 s simulation as a function of the number of recruited excitatory synapses (Fig 4B). Excitatory synapses were targeted at the distal dendrites and stimulated at 5 Hz from a Poisson distribution. By adding increasing numbers of distally targeted inhibitory synapses (also at 5 Hz) the input-output function could be shifted, or offset, to the right, particularly when $[Cl^-]_i$ was held constant during the simulations (Fig 4B, left). As has been previously described, distal inhibition did not alter the gain or maximum firing rate of the neuron. We observed that allowing $Cl^-$ to change during the course of the simulation (dynamic $Cl^-$) reduced the ability of distally targeted inhibition to offset the input-output curve (Fig 4B, right).

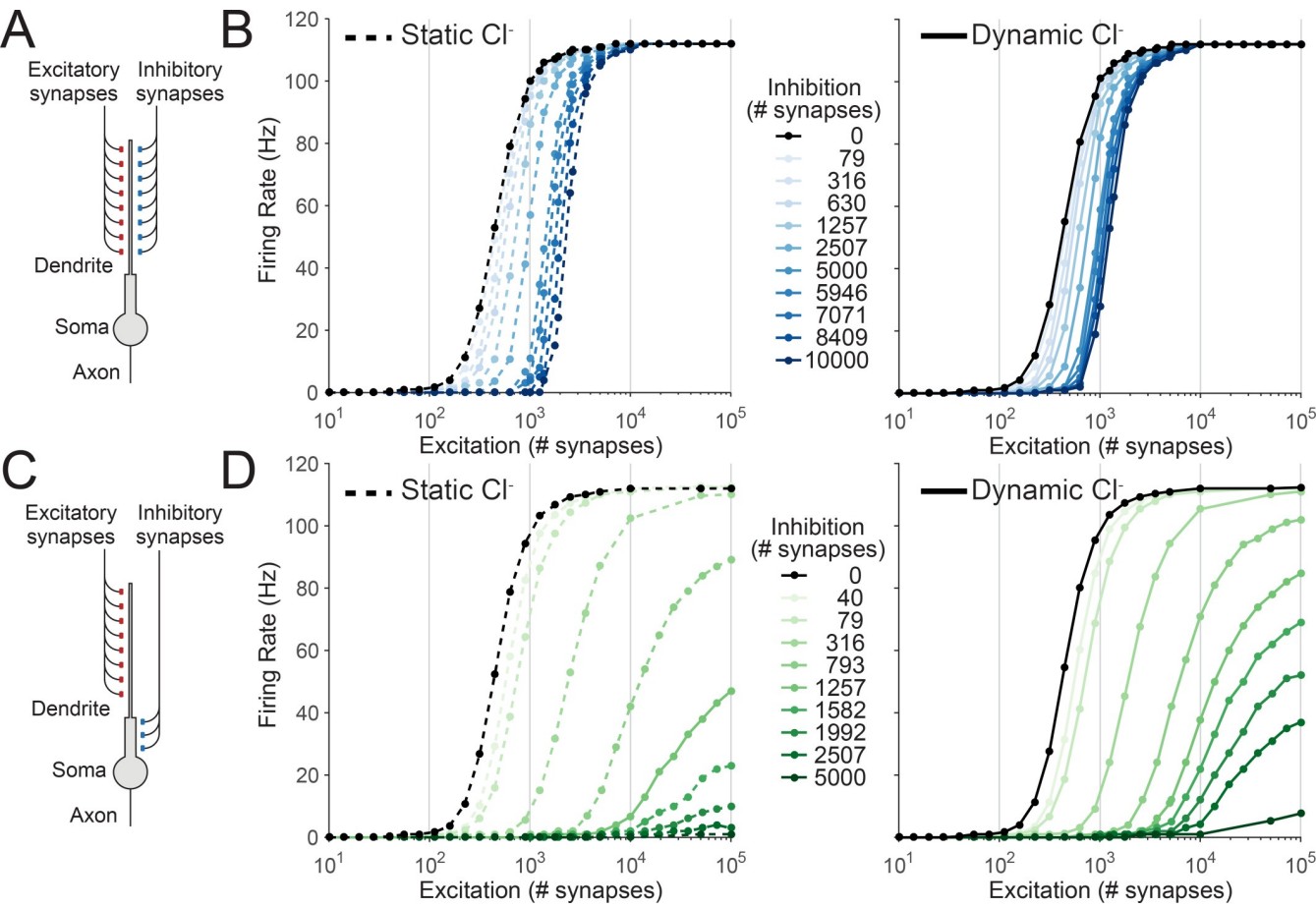

**Fig 4. The input-output function of peripherally targeted inhibition is more susceptible to the effects of chloride accumulation than proximally targeted inhibition.** (A) Schematic showing the model where both synaptic excitation and inhibition were located on the distal dendrite. Each synapse provided 5 Hz stochastic input to the neuron. (B) Left, average output firing rate over the 1 s simulation as a function of the number of excitatory synapses for different numbers of peripherally located inhibitory synapses, and where Cl⁻ was static (dashed lines). Each dot represents the result from 1 simulation (averaged over 3 trials). Increasing the number of inhibitory synapses (shades of blue) offset the neuronal input-output curve to the right (a subtractive operation on the input-output function). Right, allowing [Cl⁻]ᵢ to vary in the simulations meant that increasing the number of inhibitory synapses had a reduced ability to offset the input-output curve. (C) Schematic, showing a slightly altered version of the model with synaptic inhibition located on the proximal dendrite. (D) Left, as in 'B' but inhibition located proximal to excitatory input, i.e. peri-somatically. With static Cl⁻, increasing the number of inhibitory synapses (shades of green) generated divisive gain modulation by both offsetting the threshold and reducing the maximum firing rate of the neuron. Right, output firing rates for the same number of inhibitory synapses were moderately altered with dynamic Cl⁻, yet with sufficient numbers of inhibitory synapses complete suppression of output could still be achieved.

In contrast, simulations with inhibition shifted to a more proximal location (Fig 4C) resulted in inhibition divisively modulating input-output function (Fig 4D). Whilst accounting for dynamic Cl⁻ affected the input-output curves in this condition (e.g. with 2507 inhibitory synapses), the effectiveness of proximal inhibition for divisively modulating the input-output curve remained intact for a broader range of inhibitory synapses than when inhibition was distally located (contrast Fig 4B with Fig 4D). These simulations reveal that Cl⁻ dynamics affect the neuronal input-output function, particularly when inhibition is targeted distally.

## Chloride loading causes rapid degeneration of signalling

A neuron's input-output function is typically construed as a relatively stable property. However, as demonstrated in Fig 2, continuous synaptic input causes Cl⁻ accumulation and a

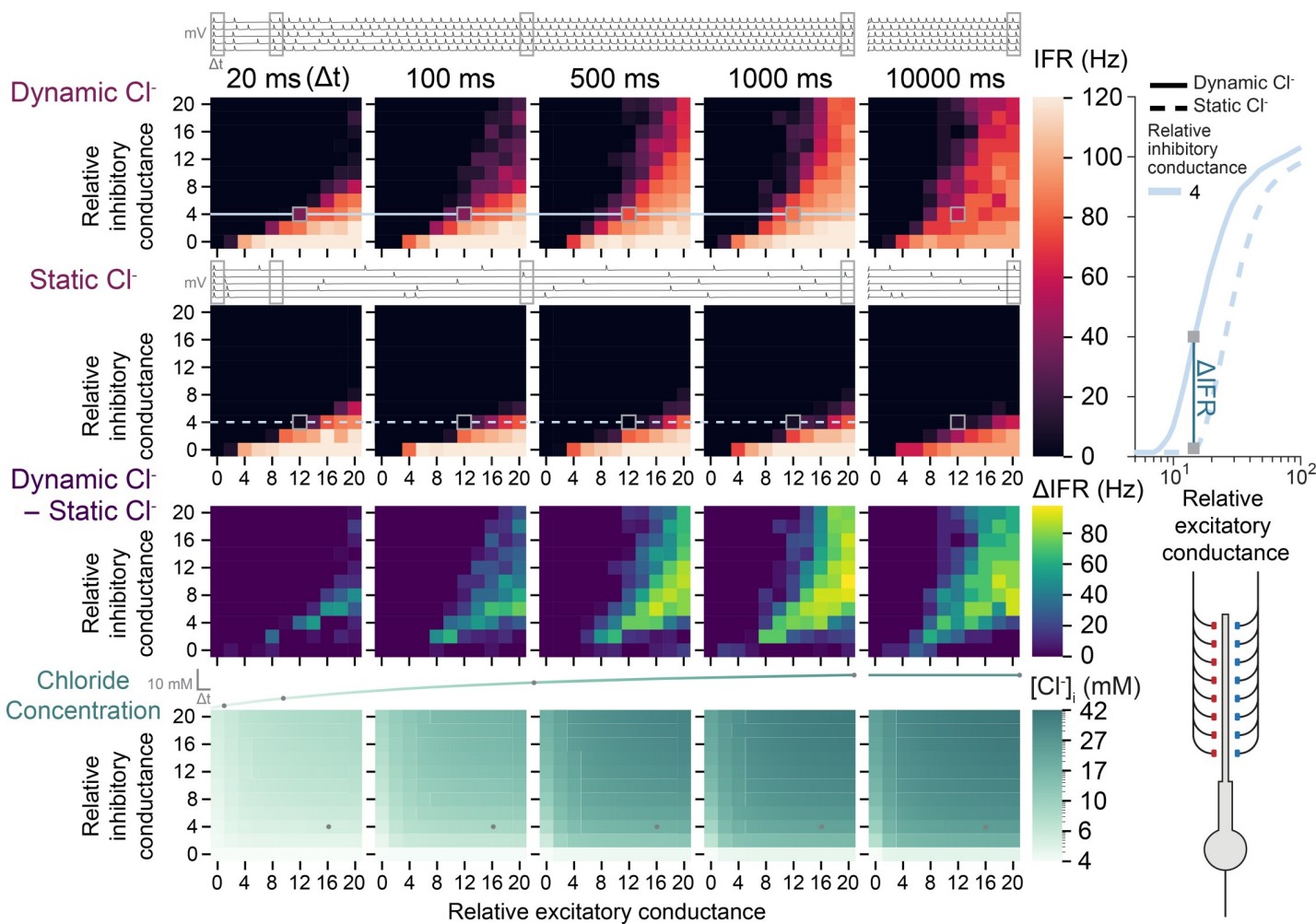

**Fig 5. Chloride accumulation has a progressive degenerative effect on the input-output function of neurons.** Neuronal input-output curves from instantaneous firing rate (IFR) represented as heatmaps using synapses with continuous fluctuating conductances instead of discrete inputs. Top, example traces of neuronal membrane potential showing action potential firing over 1 s (and at 10 s) from 5 repeated simulation runs (grey-bordered squares in the heatmaps). The IFR of the neuron was taken at 20 ms, 100 ms, 500 ms 1000 ms and 10000 ms using a backward integration window (Δt) of 20 ms as an average of the 5 simulation runs. Input-output curves represented as heatmaps of IFR for relative excitatory (different columns in an individual heatmap) and inhibitory (different rows in an individual heatmap) conductances. The top row of heatmaps is where simulations were conducted with dynamic Cl⁻. The second row is for identical simulations with static Cl⁻. The third row is the difference in IFR (ΔIFR) between the two. The bottom row is the [Cl⁻]ᵢ at the time point for that panel (a logarithmic scale was used to visualise the small changes at 20 ms). Right inset, example input-output curve for simulations with (solid trace) and without (dashed trace) dynamic Cl⁻ over 1 s. Note how differences in IFR and input-output curves emerge relatively rapidly over 1 second indicating a clear and progressive effect of dynamic Cl⁻ on signalling.

resultant shift in EGABA, which could progressively affect the input-output function over relatively short timescales. To test this idea, we repeated similar simulations to those described in Fig 4 but with synaptic drive modelled as persistently fluctuating conductances and a distal dendritic diameter of 1.0 μm. Instead of measuring the average firing rate over the entire simulation, we instead calculated the instantaneous firing rate in 20 ms time bins. This allowed us to construct input-output curves as a function of time as the simulations progressed.

We then performed multiple simulations calculating the instantaneous firing rate over time with different levels of excitatory conductances (Fig 5, x-axis of heat maps) and inhibitory conductances (Fig 5, y-axis of heat maps) with [Cl⁻]ᵢ as a dynamic variable (Fig 5, upper row) or held static (Fig 5, second row). By calculating the difference between the dynamic and static Cl⁻ conditions (Fig 5, third row), and measuring distal dendritic [Cl⁻]ᵢ within the various time

bins (Fig 5, bottom row), we were able to observe how Cl⁻ dynamics affect the input-output curves over time. We noticed that for simulations where neurons received appreciable excitatory and inhibitory synaptic drive, even after as little as 100 ms following the start of the simulation, obvious differences in instantaneous firing rate were evident. That is to say, dendritic inhibition's control over the neuronal firing rate was progressively compromised over time (Fig 5, lower row). However, the change in IFR (ΔIFR Fig 5, third row) was mostly stable after 1 s, having little further change at 10 s (Fig 5, rightmost panel). For cases where inhibitory and excitatory input was finely balanced, Cl⁻ dynamics could result in firing rate changes even within 20 ms. In general, however, these simulations demonstrate that at very short durations (<20 ms) of synaptic drive, Cl⁻ dynamics do not typically affect the input-output function of neurons, but if this synaptic input continues, changes in $[Cl^-]_i$ progressively affect the neuronal input-output curve, with the potential for large differences in instantaneous firing rate to emerge over one second.

## Quantification of the impact of chloride dynamics in models

In order to determine precisely how Cl⁻ dynamics affect the neuronal input-output function for the case of distally targeted inhibition, we established what we term the "chloride index". This measure quantifies the extent to which including Cl⁻ dynamics affects the input-output function of a neuron as compared to a condition where $[Cl^-]_i$ is held constant. That is, the "chloride index" is defined as the input-output function shift in a model where Cl⁻ is allowed to evolve dynamically during the simulation (dynamic Cl⁻) as compared to a model with static Cl⁻, relative to the case of no inhibition, as is shown schematically in Fig 6A. This shift is measured from the relative excitatory conductance at the half-max firing rate, $x_{50}$, for a given amount of inhibition $i$.

$$\text{chloride index} = \frac{b-a}{b} = 1 - \frac{a}{b}$$

Where

$$a = x_{50,i}^{\Delta Cl^-} - x_{50,i=0}$$

and

$$b = x_{50,i}^{Cl^-} - x_{50,i=0}$$

$x_{50,i}^{\Delta Cl^-}$ is the relative excitatory conductance at the half-max firing rate where Cl⁻ is dynamic, $x_{50,i}^{Cl^-}$ is the relative excitatory conductance at the half-max firing rate where Cl⁻ is static and $x_{50,i=0}$ is the relative excitatory conductance at the half-max firing rate where there is no inhibition.

In these simulations, synaptic drive was modelled as persistently fluctuating conductances over the full 1 s of each simulation. Firing rates were taken as the average firing rate over the full 1 s simulation. Similarly to the case in Figs 4 and 5, increased inhibitory drive to the distal dendrite offset the input-output curve to the right, with this being reduced by incorporating dynamic Cl⁻. This effect was quantified using the chloride index described above (Fig 6A).

We then repeated these simulations whilst adjusting the model in various ways which were predicted to alter the dynamics of Cl⁻ accumulation. First, we systematically adjusted the strength of KCC2 ($P_{KCC2}$), whilst measuring the chloride index as well as EGABA in the distal dendrite (Fig 6B). As demonstrated in Fig 6B, simulations with increased distally targeted inhibition, and in cases where KCC2 activity was reduced, had a greater chloride index because

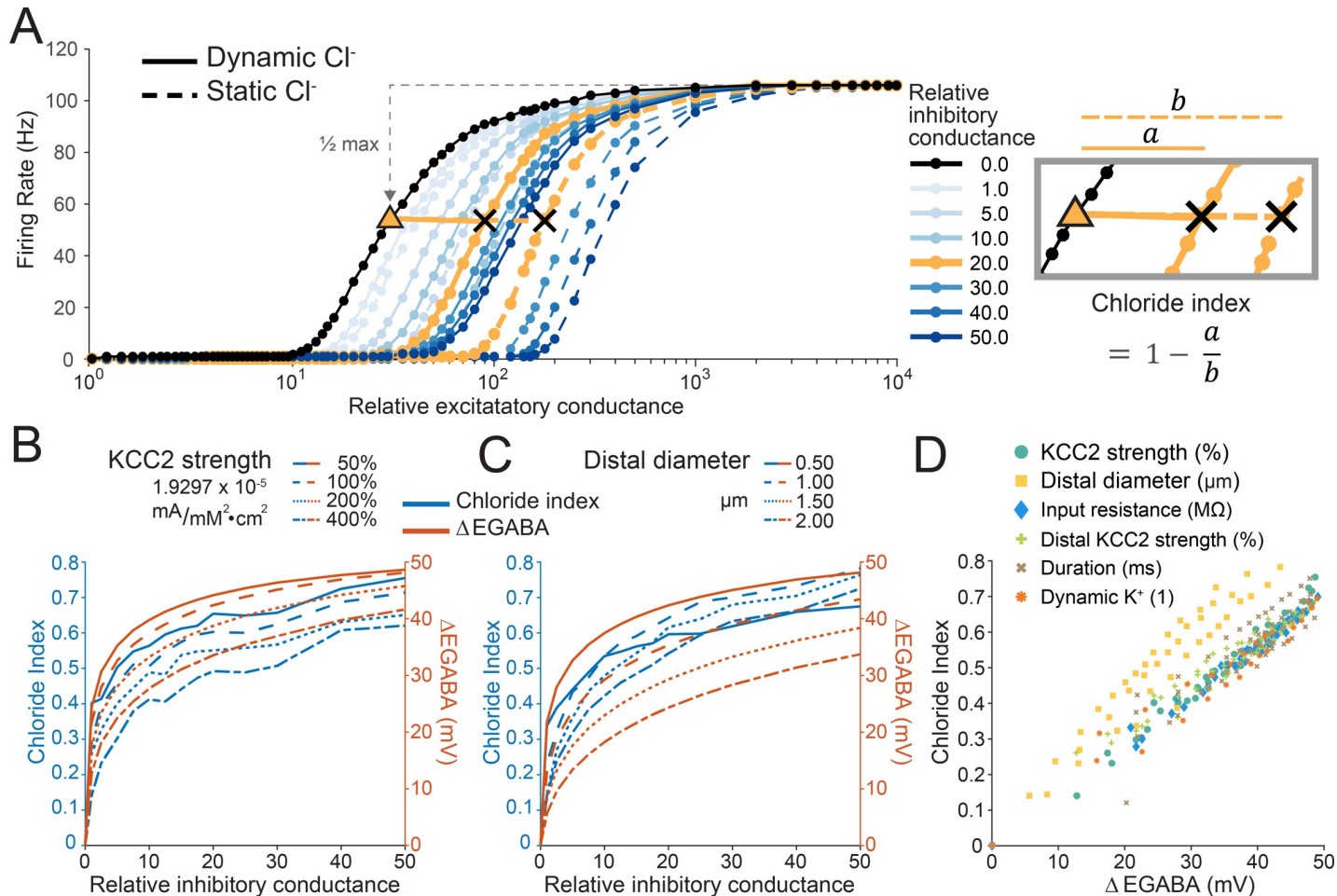

**Fig 6. Chloride dynamics shift the neuronal input-output function via changes to EGABA.** (A) As in Figs 4 and 5, increasing peripherally targeted inhibition (shades of blue) offset the input-output function, with this effect reduced when Cl⁻ was modelled as dynamic over the course of 1 s simulations (solid lines) as compared to when it was static (dashed lines). The effect of Cl⁻ dynamics on the neuronal input-output curve was quantified using the "chloride index". Orange traces, and equation (right) demonstrates how the chloride index is calculated using the example of a simulation with a relative inhibitory conductance of 4. Higher Cl⁻ indices (closer to 1) indicate a reduced effect of inhibition due to Cl⁻ being modelled as dynamic instead of static. (B) Chloride index (blue, traces) and the change in dendritic EGABA (ΔEGABA, orange traces) after a 1 s simulation for different KCC2 pump strengths ($P_{KCC2}$, 100% = 1.9297 x 10⁻⁵ mA/mM²/cm²) as a function of differing levels of peripherally targeted inhibition. Simulations with increased amounts of inhibition resulted in larger shifts in chloride index and ΔEGABA. (C) As in 'B' but with $P_{KCC2}$ held at the default level (100%, 1.9297 x 10⁻⁵ mA/mM²/cm²) and the diameter of the distal dendrite varied between 0.5 and 2 μm. Small dendritic diameters resulted in larger shifts in chloride index and ΔEGABA during the simulation. (D) Chloride index versus ΔEGABA for all the simulation runs in 'B' and 'C' as well as manipulations where we altered input resistance (S1A Fig), distal KCC2 strength (S1B Fig), the duration of the simulation (S1C Fig) and also modelled K⁺ as being dynamic (S1D Fig). The relationship between chloride index–the functional result of accounting for Cl⁻ dynamics–and ΔEGABA is directly proportional and independent of underlying neuronal properties such as KCC2 strength or diameter (r = 0.94281, p < 0.00001, Pearson correlation, N = 292 simulations).

these conditions promote Cl⁻ accumulation. In these simulations, we also measured average EGABA in the distal dendrite at the end of the simulations. We noted a close association between dendritic EGABA (Fig 6B, orange) and the chloride index (Fig 6B, blue). We then repeated these simulations, but instead of adjusting the strength of KCC2, we held KCC2 strength constant and systematically adjusted the diameter of the distal dendrite. Reducing the diameter of the distal dendrite (thereby reducing the volume of this section) also increased the chloride index for any given simulation (Fig 6C). Again, we noticed a close correspondence between the chloride index and EGABA in the distal dendrite. In addition, previous research has found that Cl⁻ accumulation depends on a neuron's leak conductance [42], that neurons have inhomogeneous KCC2 expression, with more in distal dendrites [33,35,36], and [K⁺]

changes can affect KCC2's $[Cl^-]_i$ regulation [43]. Therefore, other manipulations performed included: systematically adjusting the neurons input resistance by globally changing ionic leak conductances while holding the ratio of $K^+$:$Na^+$:$Cl^-$ conductances constant (S1A Fig), specifically enhancing KCC2 strength in the distal dendrite relative to the rest of the neuron (S1B Fig), changing the duration of the simulation itself between 500 ms to 10 000 ms (S1C Fig) and adding dynamic $K^+$ (see Methods, S1D Fig). When including all simulations, we found a very close, almost linear, relationship between the change in EGABA during the simulation in the distal dendrite and the chloride index (r = 0.94281, p $< 10^{-140}$, Pearson correlation, N = 292 simulations, Fig 6D). This clearly demonstrates that regardless of the underlying mechanism, the effect of $Cl^-$ dynamics on the input-output function of a neuron can be predicted from the change in EGABA.

## Discussion

The transmembrane $Cl^-$ concentration is critically important for regulating the properties of fast synaptic inhibition in the brain. Here we have used computational models of morphologically simplified, multi-compartment neurons that incorporated experimentally parametrised neuronal $Cl^-$ extrusion in order to provide a framework for investigating the impact of $Cl^-$ dynamics on the input-output properties of neurons. We found that continuous excitatory or inhibitory synaptic input can result in subcellular spatial differences in $[Cl^-]_i$, and that shifts in $[Cl^-]_i$ during synaptic input can affect the input-output properties of neurons.

The idea that $[Cl^-]_i$ can differ spatially within a particular neuron, depending on the subcellular compartment concerned, is important for understanding the effect of spatially targeted synaptic inhibition [10], as well as the consequences of the latest optogenetic silencing strategies which utilise light-activated $Cl^-$ channels [44]. Many interacting variables determine resting $[Cl^-]_i$ in neurons [20,22], but subcellular differences $[Cl^-]_i$ are typically attributed to differences in the activity of cation-chloride cotransporters such as KCC2 or NKCC1. For example, these may be differentially active across the somato-dendritic axis of different cell types with important implications for the cell-type specific properties of spatially targeted inhibitory synaptic transmission [45]. In support of previous work [20,21], we show that the relative amounts of inhibitory and excitatory drive can also produce spatial variations in $[Cl^-]_i$ despite uniform $Cl^-$ extrusion capacity. Notably, we show that even dendritic excitatory synaptic input alone is sufficient to drive shifts $[Cl^-]_i$ by changing the driving force for $Cl^-$ flux via tonic $Cl^-$ channels. This framework will be useful for experimentalists interpreting data collected using the latest tools to observe subcellular $[Cl^-]_i$ *in vivo* [46], where different activity states will likely affect synaptic drive onto neurons.

Previous experimental and computational work has demonstrated that for a given $Cl^-$ load via activated synaptic $GABA_ARs$, $Cl^-$ accumulation occurs more readily in smaller volume peripheral dendrites than more proximal dendritic and somatic compartments [16,18,19,21]. The effect of this on the input-output properties of neurons, however, has not previously been explored. We found that accounting for $Cl^-$ dynamics has a more pronounced effect on dendritically targeted inhibition than proximally located inhibition. This is for several intuitive reasons. Firstly, inhibitory inputs occur on smaller volume structures, which means that for a given $Cl^-$ load, there is a greater shift in $[Cl^-]_i$ and EGABA. Secondly, inhibitory inputs are co-located with excitatory inputs so the driving force for $Cl^-$ influx is typically greater for the combined excitatory and inhibitory synaptic drive. The result is that the effectiveness of distal inhibition is more susceptible to the weakening effect of $Cl^-$ accumulation than proximal inhibition and may be part of the reason why the bulk of hippocampal and cortical pyramidal neuron inhibitory synapses are typically located on their proximal dendrites and somata [31].

In our models, including $Cl^-$ dynamics reduced the ability of distally targeted inhibition to offset the neuronal input-output function. That is, it reduced the ability of distally targeted inhibition to perform a subtractive operation. And because greater inhibitory synaptic drive enhances $Cl^-$ accumulation and counterintuitively weakens inhibition further, it is not possible to overcome this inhibitory weakening by increasing the strength of distal inhibitory synaptic drive. That is, beyond a certain level, simply increasing GABAergic synaptic activation had no additional "inhibitory" effect. We also show that this perturbation of the input-output function happens progressively over time. Particularly under conditions where inhibitory and excitatory input is finely balanced, $Cl^-$ dynamics can rapidly and substantially alter instantaneous firing rates. This underscores the fact that even for neurons with a fixed structure of synaptic input, dynamic $Cl^-$ progressively alters the neuronal input-output function. This supports previous work, using single-compartment neurons, that demonstrates how, and under which paradigms, $Cl^-$ dynamics can degrade neuronal coding [24]. This also confirms previous modelling studies that show how shifts in EGABA compromise the inhibitory control of firing rates and, in turn, suggest the targeting of $Cl^-$ extrusion mechanisms and $GABA_A$ receptors as part of therapies for neuropathic pain and spinal cord injury [47,48]. By using a measure we termed the "chloride index" we were able to demonstrate that across various conditions (i.e. different KCC2 pump strengths or dendritic diameters) the shift in dendritic EGABA during synaptic input predicts how $Cl^-$ dynamics affect the neuronal input-output function.

Whilst we have demonstrated that $Cl^-$ dynamics compromise the ability of distally targeted inhibition to control neuronal output in the form of action potential generation at the soma and axon, in some cell types, particularly pyramidal neurons, it is clear that dendrites host active conductances and that non-linear input summation and integration also occurs within the dendritic tree itself [49,50]. Given this, it is likely that an important function of peripherally targeted inhibition is to control local non-linear summation of excitatory input, which may be more resistant to the effects of $Cl^-$ dynamics. Future work will need to determine how $Cl^-$ dynamics affects, and is affected by, active dendritic conductances.

We have shown how changes in $[Cl^-]_i$ have significant importance for the control of neuronal output. This is predicted to have important implications for the regulation of network excitability under both physiological and pathological conditions [51]. As we demonstrate here, for the case of distally targeted dendritic inhibition, even small activity-driven changes in dendritic $[Cl^-]_i$ can affect neuronal output, with likely implications for network activity. One notable example is the cortical up-state, when both excitatory and inhibitory synaptic drives on to pyramidal neurons are markedly elevated [52]. Our simulations predict that over the course of these events, which typically last between 0.5–1 s, intracellular $Cl^-$ may increase significantly, with a pronounced effect on the input-output function of these cells, and also on the flow of information through the cortical network [53]. As a pathological example, the intense, simultaneous excitatory and inhibitory input received by pyramidal neurons during the initiation or spread of epileptic seizures results in severe $Cl^-$ accumulation that initially weakens inhibition before resulting in paradoxical GABAergic excitation that perpetuates epileptiform activity [51,54,55]. Together, our results highlight the importance of accounting for changes in $Cl^-$ concentration in theoretical and computer-based models that seek to explore the computational properties of dendritic inhibition.

## Methods

### Experimental procedures

Rat organotypic hippocampal slice cultures were prepared using a method similar to that described by Stoppini and colleagues [56]. All experiments using animal tissue were in

accordance with regulations from United Kingdom Home Office Animals (Scientific Procedures) Act. 7 day old male Wistar rats were killed and the brains extracted and placed in cold (4˚C) Geys Balanced Salt Solution (GBSS), supplemented with D-glucose (34.7 mM). The hemispheres were separated and individual hippocampi were removed and immediately sectioned into 350 μm thick slices on a McIlwain tissue chopper. Slices were rinsed in cold dissection media, placed onto Millicell-CM membranes and maintained in culture media containing 25% EBSS, 50% MEM, 25% heat-inactivated horse serum, glucose, and B27 (Invitrogen). Slices were incubated at 36˚C in a 5% $CO_2$-humidified incubator before transfection. Neurons were biolistically transfected after 5–6 days *in vitro* using a Helios Gene Gun (120 psi; Bio-Rad) with pLenti-hSyn-eNpHR3.0-EYFP (eNpHR3.0 fused to EYFP and driven by the human synapsin I promoter, generously provided by the Deisseroth lab, Stanford University). 50 μg of target DNA was precipitated onto 25 mg of 1.6 μm diameter gold microcarriers and bullets generated in accordance with the manufacturer's instructions (Bio-Rad). At the time of recording, transfected neurons were equivalent to postnatal day 14–17. Previous work has shown that the pyramidal neurons in the organotypic hippocampal brain slice have mature $Cl^-$ homeostasis mechanisms, as evidenced by their hyperpolarizing EGABA [26,27] and that KCC2 is the major active $Cl^-$ transporter in these neurons as EGABA is affected by KCC2-blocking drugs, but not by NKCC1-blocking drugs [28].

Hippocampal slices were transferred to the recording chamber and continuously superfused with 95% $O_2$–5% $CO_2$ oxygenated artificial cerebrospinal fluid (ACSF), heated to 30˚C. The ACSF was composed of (in mM) NaCl (120), KCl (3), $MgCl_2$ (2), $CaCl_2$ (2), $NaH_2PO_4$ (1.2); $NaHCO_3$ (23); D-Glucose (11) and the pH was adjusted to be between 7.38 and 7.42 using NaOH (0.1 mM). Glutamatergic receptors and $GABA_BRs$ were blocked with kynurenic acid (2 mM) and CGP55845 (5 μM). Neurons within the pyramidal cell layer of the CA1 and CA3 regions of the hippocampus were visualized under a 20x water-immersion objective (Olympus) and targeted for recording. Patch pipettes of 3–7 MΩ tip resistance were pulled from filamental borosilicate glass capillaries with an outer diameter of 1.2 mm and an inner diameter of 0.69 mm (Harvard Apparatus Ltd, Hertfordshire, UK), using a horizontal puller (Sutter). For gramicidin perforated patch-clamp recordings[57], pipettes were filled with a high KCl internal solution whose composition was (in mM): KCl (135), $Na_2ATP$ (4), 0.3 $Na_3GTP$ (0.3), $MgCl_2$ (2), and HEPES (10). Gramicidin (Calbiochem) was dissolved in dimethylsulfoxide to achieve a stock solution of 4 mg/ml. This was then diluted in internal solution immediately prior to experimentation (10 min before attaining a patch) to achieve a final concentration of 80 μg/ml. The resulting solution was vortexed for 1 min, sonicated for 30 s and then filtered with a 0.45 μm pore cellulose acetate membrane filter (Nalgene). The osmolarity of internal solutions was adjusted to 290 mOsm and the pH was adjusted to 7.38 with KOH.

$GABA_ARs$ were activated by delivering short 'puffs' of GABA (100 μM) in the presence of glutamate receptor blockers and $GABA_BR$ blockers (see above). The agonist was applied via a patch pipette positioned close to the soma and connected to a picospritzer (20 psi for 20 ms; General Valve). To measure $[Cl^-]_i$ recovery following a photocurrent-induced $[Cl^-]_i$ load, it was important to estimate EGABA, and hence $[Cl^-]_i$, from single $GABA_AR$ currents. To achieve this, the resting EGABA and $GABA_AR$ conductance (gGABA) were calculated before each experiment and these values were then used to estimate EGABA for a single $GABA_AR$ current by assuming a consistent gGABA across GABA puffs and solving the equation $I_{GABA_A} = g_{GABA}(\text{Holding potential} - E_{GABA})$. EGABA was converted to $[Cl^-]_i$ using the Nernst equation and assuming a $GABA_AR$ $Cl^-$ to $HCO_3^-$ permeability ratio of 4 to 1, with $[HCO_3^-]_i$ = 10 mM and $[HCO_3^-]_o$ = 25 mM. Photoactivation of NpHR to generate 15 s $Cl^-$ photocurrents

was achieved via a diode-pumped solid state (DPSS) laser, with a maximum output of 35 mW and a peak at 532 nm (Shanghai Laser Optic Century). The laser was attenuated via a 5 % neutral density and coupled to a 1000 μm diameter mulitimode optic fiber via a collimating lens (Thorlabs).

## Modelling

A multi-compartmental, conductance-based neuron model was constructed in NEURON [58]. Unless The neuron was constructed as a soma extending to a short (50 μm length), thick (2 μm diameter), 'proximal' dendrite connected to a long (500 μm length), thin (0.5 μm diameter unless stated otherwise), 'distal' dendrite as in [7,37]. Spikes were recorded from the end of the 500 μm long axon (0.1 μm diameter). The neuronal morphology used is depicted to scale in Fig 1. The membrane potential was updated at each time step according to transmembrane ion currents from channels and pumps, taking into account the membrane capacitance, together calculated as $C_m \frac{dV_m}{dt} = -\sum I$. Some simulations used NEURON's variable time step integrator, 'CVODE', but the default time step of 0.025 ms was typically used.

Synapses were modelled as either conductance-based receptors receiving input from independent stochastic Poisson processes providing an input frequency ('f-in') or as persistent fluctuating conductances ('gclamp'). The reversal potentials of excitatory and inhibitory synapses matched those in [7]: $E_{excitation}$ = 0 mV; $E_{inhibition}$ = -74 mV. Excitatory f-in synapses were modelled as mixed AMPA and NMDA receptors, similar to [59], using dual-exponential kinetics with NMDA $Mg^{2+}$ dependence on membrane voltage [60–62] (see Table 2). Inhibitory f-in synapses were modelled as $GABA_A$ receptors based on the simple conceptual model of transmitter-receptor interaction where transmitter binds to a closed receptor channel to produce an open channel with a forward binding rate α and backward unbinding rate β [63](see Table 2). The $g_{max}$ values for mixed AMPA+NMDA and $GABA_A$ receptors were 1 nS and 350 pS, respectively, and strength increased by adding synapses. Each f-in synapse received input at a particular mean frequency with a standard deviation of $\sqrt{mean}$ with spike time intervals sampled from a negative exponential distribution [58]. Gclamp synapses, modelled as in [7,37], had their strength increased by multiplying their baseline passive conductance, 0.0001 μS, by a factor ('relative conductance'). Fluctuations in conductance, 'noise', was set as a coefficient of variance of 0.1. Excitatory synapses were placed on the distal dendrite. Inhibitory synapses were placed either on the proximal or distal dendrite. The number of synapses varied by experiment but were evenly distributed along the respective dendrite.

$GABA_A$ receptors, regardless of f-in or gclamp, were modelled as being selectively permeable to both $Cl^-$ and $HCO_3^-$ ions (4:1 ratio) and used the Ohmic formulation for current $I_{GABA_A} = I_{Cl^-} + I_{HCO_3^-}$, where $I_{Cl^-} = \frac{4}{5} g_{GABA}(V_m - E_{Cl^-})$ and $I_{HCO_3^-} = \frac{1}{5} g_{GABA}\left(V_m - E_{HCO_3^-}\right)$. Where applicable, the reversal potential for chloride was updated throughout the simulation using the Nernst equation, $E_{Cl^-} = \frac{R \cdot T}{F} \ln\left(\frac{[Cl^-]_i}{[Cl^-]_o}\right)$. The reversal potential for $HCO_3^-$ was held constant, and EGABA calculated as $E_{GABA} = \frac{4}{5} E_{Cl^-} + \frac{1}{5} E_{HCO_3^-}$.

The axon had channels with Hodgkin-Huxley style kinetics, as in [37], governing the sodium ($Na^+$) channel, $I_{Na^+} = m^3 h\, g_{NA^+}(V_m - E_{Na^+})$; potassium ($K^+$) channel, $I_{K^+} = n\, g_{K^+}(V_m - E_{K^+})$; potassium M-current responsible for the adaptation of firing rate and the afterhyperpolarization, $I_M = m\, g_M(V_m - E_{K^+})$; and leak conductances fitted for an input resistance of 365 MΩ with a $K^+$:$Na^+$:$Cl^-$ ratio of 1:0.23:0.4. Leak conductances were also in the soma and dendrites and, with KCC2, set the resting $[Cl^-]_i$ to 4.25 mM. Unless stated as 'static chloride', $[Cl^-]_i$ was allowed to vary ('dynamic $Cl^-$'). Transmembrane $Cl^-$ fluxes due to $Cl^-$ currents through $GABA_A$Rs, KCC2 co-transporter (as in Fig 1), as well as changes due to

**Table 2. Constants, parameters, and default steady state values for variables.**

| Symbol | Value | Description |
|---|---|---|
| Constants | | |
| F | 96485.33 C mol$^{-1}$ | Faraday constant |
| R | 8.31446 J K$^{-1}$ mol$^{-1}$ | Universal gas constant |
| T | 310.15 K | Absolute temperature (= 37˚C) |
| Parameters | | |
| $\tau_{riseAMPA}$ | 0.2 ms | AMPA rise time constant [59] |
| $\tau_{decayAMPA}$ | 1.7 ms | AMPA decay time constant [59] |
| $\tau_{riseNMDA}$ | 2.04 ms | NMDA rise time constant [62] |
| $\tau_{decayNMDA}$ | 75.2 ms | NMDA decay time constant [62] |
| $\alpha_{GABA}$ | 5 mM$^{-1}$ ms$^{-1}$ | GABA$_A$ receptor binding rate [63] |
| $\beta_{GABA}$ | 0.18 ms$^{-1}$ | GABA$_A$ receptor unbinding rate [63] |
| $P_{KCC2}$ | 0.001 mM$^{-1}$ s$^{-1}$<br>1.9297 x 10$^{-5}$ mA mM$^{-2}$ cm$^{-2}$ | Pump strength for potassium-chloride cotransporter 2 (KCC2) |
| $[K^+]_i$ | 140 mM | Internal potassium concentration |
| $[Na^+]_i$ | 10 mM | Internal sodium concentration |
| $[HCO_3^-]_i$ | 12 mM | Internal bicarbonate concentration |
| $[Cl^-]_o$ | 135 mM | External chloride concentration |
| $[K^+]_o$ | 4.0 mM | External potassium concentration |
| $[Na^+]_o$ | 140 mM | External sodium concentration |
| $[HCO_3^-]_o$ | 23 mM | External bicarbonate concentration |
| Variables (default steady state values) | | |
| ECl$^-$ | -92.42 mV | Reversal potential for chloride |
| $[Cl^-]_i$ | 4.25 mM | Internal chloride concentration |
| EGABA | -77.41 mV | Reversal potential for GABA$_A$R |
| $V_m$ | -71.35 mV | Membrane Voltage |

longitudinal diffusion, were calculated as

$$\frac{d[Cl^-]_i}{dt} = \frac{I_{Cl^-}}{F \cdot Vol} + P_{KCC2}\left([K^+]_i \cdot [Cl^-]_i - [K^+]_o \cdot [Cl^-]_o\right) + D_{Cl^-}\frac{d[Cl^-]_i}{dx}$$

where $P_{KCC2}$ is the "pump strength" of chloride extrusion, $D_{Cl^-}$ is the diffusion coefficient for chloride in water, F is Faraday's constant, Vol is the volume of the compartment, and x is the longitudinal distance between the midpoint of compartments (see Table 2 for parameter values).

For simulations where the strength of KCC2 was adjusted (Fig 6 and S1 Fig), we varied $P_{KCC2}$. For simulations where we also modelled K$^+$ as dynamic, we set the K$^+$ current through KCC2 to be opposite that of the Cl$^-$ current such that the net current through KCC2 was zero [20]. Additionally in dynamic K$^+$ simulations, previously described leak channels were replaced with a passive leak conductance ('gpas') and tonic currents that maintained the initial $[Cl^-]_i$ and $[K^+]_i$ (see Table 2).

The firing rate of a neuron was determined by counting the number of action potentials at the tip of the axon over a 1000 millisecond period. Where applicable, the instantaneous firing rate (IFR) was calculated as

$$IFR(t) = \frac{1}{K\Delta t}\sum_{k=1}^{K} n_k^{sp}(t)$$

where $K$ (number of trials) = 5, $\Delta t$ (time bin) = 20 ms, and $n_k^{sp}(t)$ was the number of spikes recorded in trial $k$ over the time window $[t-\Delta t, t]$.

## Supporting information

**S1 Text. Converting KCC2 "pump strength" parameter from $\frac{1}{mM\ s}$ to $\frac{mA}{mM^2\ cm^2}$.**
(DOCX)

**S1 Fig. Chloride dynamics shift the neuronal input-output function via changes to EGABA under multiple conditions.**
(PDF)

## Author Contributions

**Conceptualization:** Christopher B. Currin, Andrew J. Trevelyan, Colin J. Akerman, Joseph V. Raimondo.

**Formal analysis:** Christopher B. Currin.

**Funding acquisition:** Colin J. Akerman, Joseph V. Raimondo.

**Investigation:** Christopher B. Currin, Joseph V. Raimondo.

**Methodology:** Christopher B. Currin, Andrew J. Trevelyan, Joseph V. Raimondo.

**Project administration:** Joseph V. Raimondo.

**Software:** Christopher B. Currin.

**Supervision:** Andrew J. Trevelyan, Colin J. Akerman, Joseph V. Raimondo.

**Visualization:** Christopher B. Currin, Joseph V. Raimondo.

**Writing – original draft:** Christopher B. Currin, Joseph V. Raimondo.

**Writing – review & editing:** Christopher B. Currin, Andrew J. Trevelyan, Colin J. Akerman, Joseph V. Raimondo.

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
