## [Decision Letter · Decision Letter 0]

15 Oct 2019

Dear Dr Raimondo,

Thank you very much for submitting your manuscript 'Chloride dynamics alter the input-output properties of neurons' for review by PLOS Computational Biology. Your manuscript has been fully evaluated by the PLOS Computational Biology editorial team and in this case also by independent peer reviewers. The reviewers appreciated the attention to an important problem, but raised some substantial concerns about the manuscript as it currently stands. While your manuscript cannot be accepted in its present form, we are willing to consider a revised version in which the issues raised by the reviewers have been adequately addressed. We cannot, of course, promise publication at that time.

Your revisions should address the specific points made by each reviewer. In particular, make sure to address the points raised by reviewer #3 regarding originality of your model and novelty of your findings.

Please return the revised version within the next 60 days. If you anticipate any delay in its return, we ask that you let us know the expected resubmission date by email at ploscompbiol@plos.org. Revised manuscripts received beyond 60 days may require evaluation and peer review similar to that applied to newly submitted manuscripts.

Sincerely,

Hugues Berry

Associate Editor

PLOS Computational Biology

Kim Blackwell

Deputy Editor

PLOS Computational Biology

[LINK]

Reviewer's Responses to Questions

**Comments to the Authors:**

Reviewer #1: The paper combines experimental work and computer simulations to investigate the impact of fluctuations in Cl- concentration on neural signalling which is an important topic. I congratulate the authors for an interesting an well written paper. I like the concept of Cl- index. I however had the following questions/comments

1) The simulations last for 1 second. If I understand correctly, intracellular Cl concentration starts at a low level and then accumulates throughout the simulations. For the results to be relevant, Cl concentration must have reached some kind of steady-state or mean steady state level by the end of the simulations. How do they authors ensure that Cl concentration has converged during the 1s timespan of the simulation? Since dynamics of ionic concentrations are typically very slow, it is not clear to me a priori that 1 second is enough? I suggest the authors increase the duration of the simulations or at least provide a convincing argument for the convergence of Cl concentration.

2)Cl concentration is taken as dynamical while other ionic concentrations are taken as constant. While in some cases this can be a good approximation, this assumption is ultimatey inconsistent with assumptions of constant osmolority and the almost electroneutrality of the intracellular space. Given the importance of intracellular potassium in particular, I suggest the author consider intracellular K+ as dynamically evolvoing in their model. A reasonable guess is that the sum of intracellular Cl- and K+ concentrations would be roughly constant. This would impact KCC2 tranport by making the impact of Cl increase on the rate of transport less important. Maybe this could be done just for a few simulations at least to estimate how large would the impact of fluctuating potassium concentration would be.

3) The authors adjust parameters so the mean firing rate is 5 Hz when each of the synapse receives input at 5 Hz. This seems reasonable, and seems to come from a steady-state assumption such as every neuron in a network firing at 5Hz. However this is not satisfactorily justified or referenced in the paper.

4) In the discussion, the author refer to epilepsy. While the design of the paper is good is it stands, the impact of the paper could be increased if relvance of the results with respect to a particular pathology or physiological scenario would be more fleshed out.

Reviewer #2: In their computational study, the authors studied how chloride dynamics affects distal and proximal inhibition and how it modulates subtractive or divisive changes of the input-output function of neurons, which are driven by concurrent balanced excitatory synaptic drive. They found that spatial variations in chloride concentration can emerge despite uniform expression of chloride pumps. In addition, they observed that whereas chloride accumulation in dendrites diminishes the ability of distal inhibition to trigger subtractive shifts in the input-output curve, the ability of proximal inhibition to trigger divisive changes of the input-output curve remained largely unaffected by dynamical chloride. The methods are well described and the results are appropriately interpreted.

Major points:

* The authors show convincingly that concurrent inhibitory and excitatory synaptic drive may lead to spatial variations in chloride despite uniform expression of chloride pumps. However, experimental data show that the expression of chloride pumps may not be uniform leading to a dendro-somato-axonal gradient in EGABA with chloride concentration becoming progressively higher toward soma (Gulyas et al. Eur J Neurosci 2001, Szabadics et al. Science 2005, Khirug et al. J Neurosci 2008). The authors should explore such scenario in their model – e.g. by making the KCC2 pump strength spatially inhomogeneous with a higher expression in dendrites/soma and low/zero expression in the axon. Can enhanced KCC2 pump strength in dendrites counteract the inhibition-compromising effects of dendritic chloride accumulation?

* Line 155: “Combined with mechanisms to describe Cl- influx via both tonic and synaptic activation of GABAARs…” In Methods, the authors describe the synaptic models of phasic GABA-A responses. Did they include also tonic GABA-A receptors/conductances in their models? I could not find the description of them. They probably mean tonic chloride currents via leak chloride channels. This has to be clearly distinguished and described in the paper.

* Line 582-583: The authors report that they modelled excitatory synapses as voltage-dependent NMDA conductances. Were AMPA synapses not included? If not, why? Mixed AMPA/NMDA synaptic models would be biologically more realistic than pure NMDA inputs. Recent compartmental modelling of chloride concentration changes used stochastic AMPA/GABA-A inputs (Lombardi et al. Front Cell Neurosci 2018, Lombardi et al. IMJS 2019). Are the results of the current paper in line with this modelling or are there some important differences when NMDA synapses are included?

* Lombardi et al. IMJS 2019 showed that membrane resistance (determined by gpas) critically influences the amount of chloride changes. Is that the case in the authors’ model? Do their conclusion hold for a wide range of gpas values? (Modeling in the paper was performed only for a selected input resistance of 350 Mohm.)

* Prescott, Sejnowski and De Koninck (Mol Pain 2006) showed that inhibitory regulation of firing can be compromised by decreasing hyperpolarization and/or shunting. The authors should discuss their results in the light of this paper/ these findings.

* Figure 4 illustrates subtractive and divisive effects of distal and proximal inhibition, respectively. Subtractive inhibition should result in a rightward shift of the IO curve without affecting its slope whereas divisive inhibition should alter the slope of the IO curve. However, in the figure it seems that distal inhibition affects the slope, whereas proximal inhibition shifts the curve and only minimally affects the slope. Can authors comment on this?

Minor points:

* Figure 5: Replace “Excitatatory” by “Excitatory” (twice).

* Line 265: Add “blue”

Reviewer #3: Major concerns:

The study examines the effect of dynamic changes in [Cl]i on neuronal input-output relationship. Although there are some interesting aspects in this work, much of the results are intuitive and lack significant insight to be gained from a modeling study. For example, it is expected that the effect of inhibition on input-output function is weaker when inhibition is inter-mixed with excitation. Similarly, if [Cl]i is dynamically changing, any time delay in these changes could minimize the effects on the input-output relationship when excitation/inhibition are colocalized (Fig 4B). Most importantly, the model does not seem to reproduce the divisive effects of proximal inhibition on input-output relationship (Fig. 4D). This limits the confidence in the model itself. Model innovation is minimal. Overall, the results are not significant to improve our understanding of inhibitory effects on input-output processing in neurons.

Minor comments:

Lines 85: “inputs can perform a divisive operation by changing the slope (gain) of the input-output function, or”

What is the direction of this “change” – be specific.

Lines 97 – 99: “Intracellular Cl- concentration ([Cl-]i), and hence EGABA, can differ between

subcellular neuronal compartments, which has been suggested to result from spatial differences in

the function or expression of cation-chloride cotransporters in the neuronal membrane (10–12).”

Clarify the above statement.

Lines 110 – 113: “Whilst the vast majority of theoretical models of inhibitory signalling and neuronal computation assume static values for EGABA, previous studies have addressed the biophysical underpinnings of Cl homeostasis and ionic plasticity in neurons (20,22), the effects of neuronal morphologies on Cl accumulation (19,21,23), and how dynamic Cl- affects neural coding (24).”

Please summarize briefly what these studies suggest on Cl- dynamics rather than simply stating that Cl- dynamics has been explored.

Lines 135 – 141: Please clarify the rational for using organotypic slices as a suitable preparation to measure Cl-extrusion. These sentences are too long and should be restructured for readability.

Lines 177 – 180: The sentence begins with “Extensive evidence….”, but there are no relevant citations provided. Clarifying what type of methods and preparations revealed differential [Cl]i in those experimental studies would also be pertinent.

Line 224: It may be better to call “synaptic structure” as “synaptic configuration” here and elsewhere.

Lines 235 – 237: This sentence is too long and cryptic. Please explain clearly.

Lines 243 – 244: “predicted” must be replaced with “speculated”

Lines 202 – 222: Some of the descriptions provided here in the Figure legend (e.g., balanced input) can go in the main manuscript. Figure legends must be very brief only describing the graphic components of the figure.

It would be useful to provide a figure panel showing the time series of [Cl]i in the model in response to balanced input with different synaptic configurations.

**Have all data underlying the figures and results presented in the manuscript been provided?**

Reviewer #1: Yes

Reviewer #2: Yes

Reviewer #3: Yes

PLOS authors have the option to publish the peer review history of their article (what does this mean?). If published, this will include your full peer review and any attached files.

Reviewer #1: Yes: Nicolas Doyon

Reviewer #2: No

Reviewer #3: No

---

## [Decision Letter · Decision Letter 1]

20 Apr 2020

Dear Dr Raimondo,

Thank you very much for submitting your manuscript "Chloride dynamics alter the input-output properties of neurons" for consideration at PLOS Computational Biology. As with all papers reviewed by the journal, your manuscript was reviewed by members of the editorial board and by several independent reviewers. The reviewers appreciated the attention to an important topic. Based on the reviews, we are likely to accept this manuscript for publication, providing that you modify the manuscript according to the review recommendations.

Sincerely,

Hugues Berry

Associate Editor

PLOS Computational Biology

Kim Blackwell

Deputy Editor

PLOS Computational Biology

[LINK]

Reviewer's Responses to Questions

**Comments to the Authors:**

Reviewer #1: I feel the authors have satisfactorily answered the comments of the reviewers and that the current version of the manuscript is worthy of publication as it is.

A few minor comments:

1) At several places, when the authors write Cl- and K+, the - and + are not in exponent format. Please verify.

2) I'm still uneasy with p values such as 10^(-140). Is it different from 0 then? Is it within some numerical uncertainty? Or if the p-value is so small, can we say that something is certain? Also, just above the discussion sectio, -140 should be in exponent

3) Page 19, line 560, what are ' mature and stable Cl- homeostasis mechanisms'?

4) line 579, is it mOsm or mOsM?

5) Line 606, could the authors specify what is the meaning of (\\xi=1)?

6) Figure 1 c, the units are written as (1/mMs) which can be misleading. Would it possible to write mM^-1 s^-1 or (1/(mM\\cdot s))?

Reviewer #2: The authors have addressed all my concerns. Minor points:

* A new important experimental study on dendritic inhibition has recently been published - the authors might want to cite it: https://www.nature.com/articles/s41467-019-13533-3

* The authors performed new simulations with manipulations of ionic leak conductances and the associated input resistance (Fig. 6 & S1A). However, they don't mention the rationale or the motivation for it - namely previous results of Lombardi et al. https://www.ncbi.nlm.nih.gov/pubmed/30897846 For a better understanding of the background for these simulations, this could be mentioned at the beginning of the relevant results section.

* L. 464 - replace "stength" by "strength"

Reviewer #3: The authors have made significant changes to the manuscript since last review which has greatly improved the manuscript and satisfactorily clarified much of the reviewer concerns. However, there are a few minor points to be addressed as noted below:

1) Ref Fig 2 and other related simulations. Please provide the stdev of the input firing rates and note whether it differs for different mean input firing rates (e.g.., mean rate of 5 Hz versus 20 Hz).

2) Line 414: As per the illustration in Fig 6A inset, the authors have to spell out x50 as the relative excitatory conductance at half-maximal firing rate in the results section. Currently it seems like they call the half-maximal firing rate as x50 which is not the case.

3) Figure 3, panels A and B: Please change Y-axis label to Firing Rate (Hz).

4) Line 513: Previous work such as Venugopal S, J Neurophysiol, 2011 has to be cited.

Line 189 Remove "well" or provide multiple references

Line 206 Begin sentence with "Note that multiple candidate..."

Line 207 Remove extra period

**Have all data underlying the figures and results presented in the manuscript been provided?**

Reviewer #1: Yes

Reviewer #2: None

Reviewer #3: Yes

PLOS authors have the option to publish the peer review history of their article (what does this mean?). If published, this will include your full peer review and any attached files.

Reviewer #1: Yes: Nicolas Doyon

Reviewer #2: No

Reviewer #3: No
---

## [Editor Report · Decision Letter 2]

6 May 2020

Dear Dr Raimondo,

We are pleased to inform you that your manuscript 'Chloride dynamics alter the input-output properties of neurons' has been provisionally accepted for publication in PLOS Computational Biology.

Best regards,

Hugues Berry

Associate Editor

PLOS Computational Biology

Kim Blackwell

Deputy Editor

PLOS Computational Biology

---

## [Editor Report · Acceptance letter]

20 May 2020

PCOMPBIOL-D-19-01233R2 

Chloride dynamics alter the input-output properties of neurons

Dear Dr Raimondo,

I am pleased to inform you that your manuscript has been formally accepted for publication in PLOS Computational Biology. Your manuscript is now with our production department and you will be notified of the publication date in due course.

With kind regards,

Laura Mallard
